# A Path Analysis Model of Self- vs. Educational-Context- Regulation as Combined Predictors of Well-Being in Spanish College Students

**DOI:** 10.3390/ijerph191610223

**Published:** 2022-08-17

**Authors:** Claudia López-Madrigal, Javier García-Manglano, Jesús de la Fuente Arias

**Affiliations:** 1Institute of Culture and Society, University of Navarra, 31008 Pamplona, Spain; 2School of Education and Psychology, University of Navarra, 31008 Pamplona, Spain; 3School of Psychology, University of Almería, 04120 Almería, Spain

**Keywords:** young adults, self- vs. external-regulation, educational context, coping strategies, well-being

## Abstract

Previous literature has established the importance of personal and contextual factors in college students’ trajectories. Following the Self- vs. External-Regulation Behavior Theory (2021) and the 3P Biggs Model, the present study aimed at analyzing a structural linear model that validates the joint effect of self-regulation, educational context, age, and gender (as personal and contextual presage variables) with other meta-abilities, such as coping strategies, resilience, and positivity (process variables), and specific well-being outcomes, such as flourishing and health (product variables). A sample of 1310 Spanish college students was analyzed, aged 17 to 25, and a cross-sectional study with an ex post facto design was performed. Association and structural equation modeling (SEM) was performed using SPSS software (v.26) and AMOS (v.23). Results show that individual and contextual factors have an important role in the acquisition of psychological competencies in young adults. Self-regulation was proven to be an important meta-ability that predicts personal well-being and behavioral health outcomes. Complementarily, educational context was shown to be an external predictor of other skills, such as problem-focused strategies, and positive outcomes such as flourishing and behavioral health. Practical implications and limitations are discussed.

## 1. Introduction

An increased interest in positive trajectories in young adults has grown significantly over the last 10 years [1]. More focus has been placed on promoting a set of emotional, cognitive, and behavioral skills that prepare them for their later challenges in adult life.

Literature regarding young adults is mainly focused on college students because of the opportunities and risks that arise during this period [2]. Despite the criticism around this representativeness issue [3], the academic literature has grown towards new ways of providing students with skills and abilities that go beyond academic conceptual and theoretical learning. Metaskills related to “learning to learn” prepare them not only professionally but personally [4]. More information is needed to understand how previous personal and contextual factors influence the acquisition of these skills and their impact on well-being outcomes. The main aim of this study is to build a structural linear model to better understand the joint effect of self-regulation, educational context, age, and gender as *personal and contextual presage* predictors with other meta-abilities, such as coping strategies, resilience, and positivity (*process variables*), and its relationship with well-being outcomes, such as flourishing and health (*product variables*). This aim will be guided by the theoretical frameworks detailed below.

### 1.1. Theoretical Frameworks

*The 3P* (*Presage, Process, Product*) *Model* (Biggs, 1987). Originally developed with an academic approach, this model was designed to explore the components involved in the learning process other than the mere ability of the student. A system with three main components was described: presage, process, and product. *Presage* factors relate to the experiences and context prior to the learning experience. These factors are more stable and are divided into two categories: (1) those related to the individual and personal characteristics (e.g., personality traits); (2) those related to contextual characteristics (e.g., family background). *Process* factors relate to strategies that can mediate or influence while the process of learning takes place. These can be influenced by the presage variables, and/or influence the product. Finally, *product* factors are the outcomes obtained after learning takes place. They correspond to the quantitative or qualitative results that can be influenced either by the presage or process factors [5].

The 3P Model highlights the dynamic between the pre-existent variables with the process itself or the outcomes obtained. For this reason, this model has been replicated in multiple studies with different states of affair such as management education [6], computing education [7,8], and augmented reality teaching [9]. This model was used to guide and categorize all the variables for the present study, which will be further described. Specifically:(1)*Presage* variables: self-regulation level, educational context, age, and gender.(2)*Process* variables: coping strategies, resilience, and positivity.(3)*Product* variables: flourishing and health (physical and psychological).

The *Theory of Self-* vs. *Externally Regulated Behavioral Model* [10] is a theoretical framework that defines self-regulation as an individual and contextual factor. Self-regulated behavior (SR) as a personal characteristic refers to the level of self-directedness that every individual has, which can be described in three dimensions: (1) *Self-Regulation* (SR) refers to the active and adequate management of one’s behavior; (2) *Nonregulation* (NR) is defined as the lack of proactivity and being reactive to external inputs; (3) *Dysregulation* (DR) relates to a negative and inadequate way of behavior regulation.

The novelty of this model is that it fills the gap of understanding self-regulation only as an internal skill by proposing that external and contextual variables can also be regulators [11,12]. External regulation (ER) can facilitate or diminish the acquisition of other competencies and the directionality of the behavior. The model describes three levels: (1) *External Regulatory* (ER), where the contexts actively and positively encourage self-regulation; (2) *External Nonregulatory* (ENR), which refers to a context that does not direct the behavior nor the promotion of a self-regulated behavior; (3) *External Dysregulatory* contexts (EDR), which are those that favor inadequate and negative behavior. In this kind of context, individuals engage in activities that are not favorable for them. The present study will follow this model using the following:(1)*Self-regulated* variable (SR): self-regulation.(2)*Externally regulated* variable (ESR): educational context.

### 1.2. Self-Regulation and Educational Context as Personal and Contextual Presage Variables

Self-regulation. Young adults find themselves immersed in a constantly changing environment with multiple possibilities, where they must approach and commit to their goals in life. Achieving a skill related to self-directedness will help college students to accomplish temporary and permanent stability. Self-regulation has been defined as one of the soft skills of the 21st Century [13] and a key element for competent functioning from an early age [14]; it is considered a metacognitive variable because of the skills it embodies and its role in regulating other variables even when the context is not helpful [4,12]. Other skills involved during the self-regulation process are goal setting, self-efficacy, self-observation, self-evaluation, strategic planning, and motivational strategies [15,16,17,18,19,20]. Plenty of research supports the relationship between self-regulation and other outcomes such as subjective well-being [21,22], coping strategies [12], lower levels of academic stress [23], self-efficacy [24], and as a protective factor for risk behaviors and negative health habits [25].

Following the *Self- and External-Regulation* approach, context or previous backgrounds play an important role in the acquisition of other meta-abilities. University context is something that needs to be discussed and considered in the development of cognitive, emotional, and behavioral skills. The formative context where the student is immersed impregnates most of their academic and personal experience [26]. The external conditions where college experience takes place can promote or aggravate their well-being [27]. Anxiety, depression, and stress are linked to social determinants; however, unfortunately, external stressors are not evaluated when diagnosing mental illness [28]. Scholars have discussed that school climate enables healthy academic and personal development [29]. Within the field of Positive Psychology, evidence related to character strengths has emphasized the benefits of a value-based context where human potential is praised [30]. However, in the attempt to understand what builds positive trajectories in young adults, most studies have acknowledged individual determinants being scarce in the literature regarding educational climate.

### 1.3. The Role of Age and Gender as Presage Individual Variables

Sociodemographic variables such as *age and gender* play a key role in one´s development. This study is focused on college students—people between 18 and 23 years old. These are young adults facing multiple and critical changes socially, psychologically, and physiologically, where traditional social roles are no longer clearly established [3]. The maturing process they are immersed in makes them a vulnerable population to specific stressors. The constant dynamic changes are a breeding ground for psychological pathologies to emerge [31]. Being at an age of constant exploration and commitment, each decision has an impact on their identity formation [26]. As they transition from adolescence to adulthood, there are a lot of expectations to fulfill and new contexts where they need to cope effectively.

Gender (male and female) is also an important factor to consider when analyzing other skills. Differences have been described as to how men and women experience their college period, especially when it comes to work or relational preferences [32]. Results show that women place a higher value on romantic relationships, while men tend to be more focused on academic opportunities [33]. Correspondingly, differences have been found with other strategies such as self-regulation levels, coping, resilience, and well-being [22,34,35,36,37]. Differences have also been reported with regards to mental health and risky behaviors [38]. Gender has been described as an important predictor of well-being and a strong predictor of positive development in young adults [39]. The heterogeneity of paths and gender differences must be considered when studying young adults [40,41].

### 1.4. Coping, Resilience, and Positivity as Process Variables

Coping strategies, resilience, and positivity have been previously described as metacognitive, meta-motivational, and attitudinal constructs, respectively [4,12,42]. These constructs relate to other skills and have a wide variety of behavioral implications.

The way an individual copes with their surroundings or stressors is critical for their functioning [31]. Students need to face the academic demands of the university and the context-related changes and needs, which are defined as the “cognitive and behavioral efforts to manage specific external and/or internal demands that are appraised as taxing or exceeding the resources of the person” [43]. One of the most common distinctions is problem-focused and emotion-focused coping [43]. While one aims at eliminating or reducing the cause of stress (problem-focused), the other one aims at regulating the associated emotional response (emotion-focused) [44]; both are complementary, and the use of one over the other will depend on what needs to be tackled. Coping has been previously associated with life outcomes such as resilience [42], quality of life [45], character strengths [46], and better social functioning [47]. It has been considered a protective factor against risky behaviors and mental health issues in emerging adults [34,48].

Resilience is a highly related construct to coping strategies [4,42,49]. It refers to the ability to adapt and cope with stress, show up to adversity, and maintain positive mental outcomes [50]. While coping relates to the efforts of dealing with the stressful event after the appraisal of that event, resilience relates to the evaluation and positive response to stressors [51]. Despite being a psychological construct, it has recently been studied and extended to other disciplines such as organizational psychology [52] and sustainability [53]. Considered a meta-motivational variable [42], scholars have differentiated it into proactive (factors related to the ability to face the stressor) and reactive resilience (the endurance of adverse conditions) [54]. Overcoming distress during emerging adulthood requires a mind shift that relies on a self-motivated purpose [55]. Evidence shows that resilience has a clear relationship with psychological flourishing [56], identity capital [57], academic success [58], character strengths [59], and career adaptability [60], among many other positive outcomes. Evidence shows that along with the big five personality traits, resilience acts as a buffering effect against the perceived levels of stress and the ability to effectively manage stress during college [54].

Positivity is a construct that has been widely studied in Positive Psychology as a general disposition to view life experiences in a positive way [61]. From a developmental perspective, it has been said that it is stable from adolescence to adulthood; however, it has been proven that it is something that also increases with age [62]. From a positive perspective, it is an important predictor of well-being because of its role in life perception and attitude towards adversities [63,64]. Positivity researchers have described self-esteem, life satisfaction, and optimism as common features that relate to positivity [65]. Evidence suggests that positivity acts as a strong predictor of happiness, quality of life, and positive and negative affect [66]. Further, it acts as a protective factor against psychological problems and negative outcomes [67].

### 1.5. Well-Being Correlates as a Product Variable

Flourishing has been defined as a construct that encompasses five different dimensions: positive emotion, engagement, relationships, meaning, and accomplishment [68]. Newer definitions have added or subtracted other dimensions, and there is still no consensus on a single definition. However, as VanderWeele (2017) says, all of them can agree that flourishing is understood as a “state in which all aspects of a person´s life are good” [69]. The implications and applications of flourishing are numerous and have been measured in cross-cultural studies [70,71,72]. The numerous changes and new inputs that college students have during this period might have an impact on their way to flourish. High levels of flourishing are associated with effective learning, productivity, creativity skills, and good health [40]. Moreover, positive associations have been described with other constructs such as life satisfaction [73], grit [74], self-esteem and forgiveness [75], and self-efficacy [76], amongst others.

A parallel construct is behavioral health. According to the World Health Organization (WHO), health has been defined as a “state of complete physical, mental, and social well-being, and not merely the absence of disease or infirmity”. Behavioral health is a wide-reaching term because it refers to the impact that behaviors have on someone’s physical and mental health [77]. Both mental and physical health are extremely important during the transition to adulthood [78]. Young adults are in a developmental stage of exploration where a lot of risky behaviors take place such as substance consumption, risky sexual behaviors, or sleep privation [79]; it is also a critical period where psychological pathologies tend to arise, such as depression, anxiety, stress, eating disorders, and suicidal ideation [80,81,82,83]. Taking health into consideration is central to subjective well-being and their sense of wholeness as human beings [69].

### 1.6. Aims and Hypotheses

Previous literature has established the importance of personal and contextual factors in the processes and outcomes of young adults [4]. Therefore, the main aim of this study is to build a structural linear model that validates the joint effect of self-regulation, educational context, age, and gender (as *personal and contextual presage* variables) with other meta-abilities, such as coping strategies, resilience, and positivity (*process* variables), and at the same time on well-being outcomes, such as flourishing and health (*product* variables) (Figure 1).

To this end, the following hypotheses were established:

**H1.** 
*Self-regulation and Educational context (as regulatory presage variables) will have a positive linear relationship of association and prediction with coping strategies, resilience, and positivity (process variables).*


**H2.** 
*Self-regulation and Educational context (as regulatory presage variables) will have a positive linear relationship of association and prediction with flourishing and behavioral health (product variables).*


**H3.** 
*Age and gender will mediate variables to the relationships described above.*


## 2. Materials and Methods

### 2.1. Participants

The sample consisted of 1310 Spanish college students from 17 to 25 years old. Of the sample, 25% were men (*n* = 327) and 75% were women (*n* = 983). Participants were enrolled in Education and Psychology programs. All participants were college students from universities of different autonomous communities in Spain. Most of the sample were in their first years of college (53%, *n* = 699), whilst only 11% (*n* = 139) were in their last year.

### 2.2. Instruments

#### 2.2.1. Presage Variables

*Self-regulation.* The Spanish Short Self-Regulation Questionnaire (SSSRQ) [84]. It consists of a 17-item scale designed to measure self-regulation and its related factors. It is structured in four dimensions: goal setting, perseverance, decision making, and learning from mistakes. It has a Likert Scale response option that ranges from 1 (totally disagree) to 5 (totally agree). Self-regulation groups were grouped by clusters, obtaining three different levels: low, medium, and high. Validity and reliability measures are appropriate for the Spanish sample (α = 0.86; *χ*^2^ = 641,209; *p* < 0.001; RFI = 0.949; IFI = 0.972, TLI = 0.994, CFI = 0.992, RMSEA= 0.075).

*Educational Context.* This information was obtained from sociodemographic data. Students were from different universities in Spain, and these contexts were classified respectively according to the academic programs the students attended. The educational context was assessed as a dichotomous variable (0 = context 1; 1 = context 2); context 1 is defined by professional and scientific-oriented formative contexts, whilst context 2 is defined by value-based oriented formative contexts.

*Gender and Age.* This information was gathered from the sociodemographic data of each of the respondents. Gender was assessed as a dichotomous variable (0 = men; 1 = women). Sample ages ranged from 17 to 25 years old and were grouped into three categories: 17–19 years old (youngest group), 20–22 years old (middle group), and 23–25 years old (oldest group).

#### 2.2.2. Process Variables

*Coping strategies.* The Short Spanish version of the Coping Strategies of Stress Scale (EEC-Short) [85]. The scale consists of a 64-item scale that assesses different strategies to cope with stress. The scale has two dimensions: (D1) emotion-focused coping and (D2) problem-focused coping. Each dimension is compounded by 10 different factors that account for different coping strategies. Response options have a Likert scale that ranges from 0 (never) to 4 (always). The validity and reliability measures are adequate (α = 0.93; *χ*^2^ = 878.750; *p* < 0.001; RFI = 0.945; IFI = 0.903, TLI = 0.951, CFI = 0.903, RMSEA = 0.07).

*Resilience.* The Spanish version of the Connor Davidson Resilience Scale [86]. The original scale was developed by [87] and consists of a 25-item scale with a 5-point Likert scale that ranges from 0 (never) to 4 (almost always). It was designed to assess resilience and its main factors: tenacity, stress tolerance, control perception, tolerance to change, and spirituality. The Spanish version has adequate reliability and validity measures (α = 0.88; *χ*^2^ = 1,619,170; *p* < 0.001; RFI = 0.948; IFI = 0.922; TLI = 0.908; CFI = 0.920; RMSEA = 0.063; HOELTER = 240 (*p* < 0.05) and 254 (*p* < 0.01)).

*Positivity.* The Positivity Scale (PS) [88]. This scale consists of a 10-item self-report scale that addresses some statements related to positivity, self-esteem, optimism, and life satisfaction. Item response options range from 1 (totally disagree) to 5 (totally agree). The Spanish validation has appropriate validity (α = 0.893; *χ*^2^ = 308.992; *p* < 0.001; RFI = 0.894; IFI = 0.912; TLI = 0.923; CFI = 0.916; RMSEA = 0.085; HOELTER = 260 (*p* < 0.05) and 291 (*p* < 0.01)).

#### 2.2.3. Product Variables

*Flourishing.* The Spanish version of the Flourishing Scale (FS) [89]. The original scale consists of an 8-item scale aims at measuring a construct of well-being known as flourishing [90]. It has a 5-point Likert scale that ranges from 1 (strongly disagree) to 5 (strongly agree). The Spanish validity and reliability properties of the scale are adequate (α = 0.85; *p* < 0.000; GFI = 0.94; NFI = 0.91; TLI = 0.916; CFI = 0.95; RMSEA = 0.062).

*Behavior Health.* The Spanish version of the Student Health Inventory (Cuestionario de Salud Académica) [22]. This scale was designed to evaluate the student’s health. It is compounded by two dimensions: physical health (e.g., “I sleep well”) and psychological health (e.g., “I feel depressed by studies”). It has 8 items with a 5-point Likert scale that ranges from 1 (strongly disagree) to 5 (strongly agree). Reliability and validity values are adequate for the Spanish sample (α = 0.751; χ^2^ = 41.385; *p* < 0.001; IFI = 0.941; TLI = 0.911; CFI = 0.946; RMSEA = 0.073).

### 2.3. Procedure

This study is part of two R&D Projects (reference EDU2011-24805 (2012–2015) and PGC2018-094672-B-100 (2018–2021); www.inetas.net, accessed on 18 January 2021) from the Ministry of Science, Innovation, and Universities of Spain. Researchers approached different Spanish universities to present the aims of the project to invite the teachers to be part of the study. The data were collected through a student’s self-help online application built by the researchers to offer guidance. The average response rate of the whole questionnaire was 25 min. Students were invited to participate by their teachers; participation was completely anonymous and voluntary, and no economic compensation was obtained. This study has been approved by the Ethics Committee and the Institutional Review Boards (ref. 2018.170).

### 2.4. Data Analysis

The present is a cross-sectional study with an ex post facto design [91]. Association analyses were performed using bivariate Pearson correlations. For the prediction analyses, Structural Equation Modeling (SEM) with a Maximum Likelihood estimation was conducted to test the hypothesized model. This analysis has been described to be the most appropriate to understand the direct and mediating effects of the latent predictor variables on outcome variables [92]. The fit indices used to assess the models were the Comparative Fit Index (CFI) and the Root Mean Square Error of Approximation (RMSEA). According to [93], CFI values were considered of acceptable fit if they were equal to or above 0.90. For the RMSEA, values equal to or less than 0.05 indicate a “close fit” whilst values between 0.05 and 0.08 are considered as “reasonable fit” [94]. All analyses were performed with the IBM-SPSS statistical program (v. 26) and AMOS (v. 23).

## 3. Results

### 3.1. Association Analysis: Bivariate Correlations

Pearson’s correlation analyses are shown in Table 1.

*Presage.* Positive significant associations were found between self-regulation (SR), total coping (COP), problem-focused coping (PF.COP), resilience (RESIL), and positivity (POS). Further, positive associations were found between SR with the *product* variables of flourishing (FLO) and health (HEALTH) in its physical (PHY.H) and psychological (PSY.H) components. A negative association was found between self-regulation (SR) and emotion-focused coping (EF.COP; *r* = −0.16, *p* < 0.000).

*Process.* Positive significant associations were found between the *process* and *product* variables. Coping strategies (COP) only had a positive significant association with flourishing (FLO). Emotion-focused coping (EF.COP) was negatively associated with psychological health (PSY.H; *r* = −0.28, *p* < 0.0001), whilst problem-focused coping (PF.COP) was positively associated with flourishing (FLO), and health (HEALTH) only in its physical health component (PHY.H). Finally, positivity (POS) was positively significant with both well-being constructs: flourishing, and health in its two components.

*Product.* High coefficients of association were found amongst the *product* variables, namely, positivity (POS), flourishing (FLO), and health (HEALTH) in its two components (PHY.H; PSY.H).

### 3.2. Structural Predictive Path Analysis

The SEM analysis was established following these theoretical statements: (1) self-regulation group and educational context (*personal and contextual presage* variables) would significantly and positively predict coping strategies, resilience, and positivity (*process* variables). (2) Self-regulation and educational context (*personal and contextual presage* variables) will positively predict flourishing and health (*product* variables). (3) Coping, resilience, and positivity (*process* variables) would positively predict well-being constructs, namely, flourishing, and health in both constructs (*product* variables). (4) Age and gender, which were nonlatent variables, were expected to be predictors of all the latent variables studied (emotion-focused coping, problem-focused coping, positivity, flourishing, and behavioral health). Results below will be detailed according to these theoretical statements.

Two models were tested, with the second model being the most consistent (see Table 2). In Model 1, the predictive relations of self-regulation and educational context were tested in relation to coping strategies, resilience, and positivity. However, fit indexes were not consistent. A better and more consistent model is Model 2, in which the prediction in relation to resilience was eliminated. The final model is represented in Figure 2.

### 3.3. Standardized Direct and Indirect Effects

Following the theoretical statements previously described, direct effects showed the following: (1) The self-regulation group was a positive predictor of problem-focused coping and a negative predictor of emotion-focused coping. Educational context resulted in a significant positive predictor of problem-focused coping. (2) Both the self-regulation group and educational context were positive predictors of flourishing. Accordingly, educational context was a positive predictor of health. (3) Problem-focused coping positively predicted flourishing. Positivity positively predicted flourishing and health. Flourishing positively predicted behavioral health. (4) Age group was a positive predictor of the self-regulation group and a negative predictor of emotion-focused coping. As for gender, it was a positive predictor of problem-focused coping and a negative predictor of health. See Table 3.

The indirect effects showed the following: (1) Self-regulation level and educational context were positive predictors of positivity. (2) Self-regulation level and educational context were positive predictors of all product variables, namely, flourishing and health in its two components. (3) Emotion-focused coping was a negative predictor of flourishing and health, whilst problem-focused coping was a positive predictor of them. Complementarily, positivity was a positive predictor of health in its two components, physical and psychological. Flourishing was a positive predictor of physical and psychological health. (4) Finally, age group negatively predicted emotion-focused coping but was a positive predictor of problem-focused coping, positivity, and all the product variables. Gender, as a nonlatent presage variable, was a positive predictor of positivity, flourishing, and global health (see Table 3).

## 4. Discussion

Following the main aim of this study, a structural linear model that validates the joint effect of self-regulation, educational context, age, and gender (as *personal and contextual regulatory and presage* variables) with other meta-abilities, such as coping strategies, resilience, and positivity (*process* variables), and at the same time on flourishing and health (*product* variables) was achieved. From a general perspective, the integration of individual and contextual factors relates to the acquisition of specific psychological competencies and positive well-being outcomes in young adults.

According to the *first hypothesis* (H1), it was expected that the *personal and contextual presage* variables (self-regulation and educational context) would have a positive linear relationship of association and prediction with coping strategies, resilience, and positivity (*process* variables). Similar to what previous results have found [4,22], our results confirmed that the combination of self-regulation and educational context had a positive prediction relationship with problem-focused coping and an indirect effect on positivity. These results support previous literature in that problem-focused strategies are related to solving the problem that is causing distress, and the self-regulation process involves other skills related to problem-solving such as self-efficacy, strategic planning, and self-evaluation [16,17]. Regarding educational context, it has been discussed that external conditions are important for the adequate performance and well-being of higher education students [95,96,97]. The predictive effect of both presage and contextual variables on positivity reflects the effect that previous variables have on life perception and attitude towards adversities [63,64]. Our results support the idea that external context relates to protective factors against negative outcomes such as positivity [67].

According to the *second hypothesis* (H2), it was expected that self-regulation and educational context would have a positive linear relationship of association and prediction with flourishing and behavioral health (*product* variables). Similar to what other studies have reported, our results showed that flourishing was predicted by both variables, self-regulation and educational context [22,76]. Behavioral health was indirectly predicted by both personal and contextual presage variables but directly predicted only by educational context. This second finding is very interesting because it supports the main idea that context matters. As other scholars have pointed out, the external and previous contextual variables are important to well-being and other positive outcomes [10,11,22]. In the case of college students, the university is the educational setting where developmental tasks take place and where they need to learn skills that prepare them for later adult life [26,98,99]. As [100] mentions, the academic context can facilitate or diminish the efforts of academic success and personal development.

Finally, the *third hypothesis* (H3) predicted that age and gender would mediate the presage, process, and product relations. Concerning age, the present results show that it was a negative predictor of emotion-focused coping but a positive predictor of problem-focused coping, positivity, flourishing, and health. In line with what other researchers have established, college students are in a sensitive developmental stage, which results in a critical point to either flounder or flourish [41]. This age represents a double-sided door to achieving negative or positive outcomes that provides them with the opportunity to integrate positive lifestyles that could lead them to better well-being outcomes [101]. As for gender, being female was a positive predictor of positivity, flourishing, and global health, but a negative predictor of the health’s components (physical and psychological health). These results are similar to what other scholars have pointed out [34,36,39] in that gender is a strong predictor of positive development in young adults.

According to the present results, it seems reasonable to assume that more empirical support has been provided to the *Self-* vs. *External-Regulation Behavior Theory* model [10,102] in the educational field.

### Limitations and Future Research

While the present study sheds more light on the literature, it is not without limitations. First, the sample is not representative of the Spanish youth. This sample only addressed college students. Other occupational statuses need to be addressed, as young adults with different socioeconomic or occupational statuses are immersed in radically different contexts that might affect the present results.

Second, educational context could be better defined. The sample is not equally distributed by educational context in that only two contexts were analyzed. The universities assessed do not represent the variety of educational contexts available in Spain. Specific ideas related to improving the measurement of educational context are as follows: (1) Other formations such as 2-year college, or technical professions should be compared to see if there are any differences. (2) To consider other delimiting factors such as organizational culture, reputation indexes, extracurricular programs, the university’s involvement with the civil community, or even the students’ and teachers’ perceptions. (3) To compare other types of nonacademic contexts that relate to integral development such as personal development programs, spiritual initiatives, mental health support, or character strength projects. These are activities that are not in the curricular programs but are embedded in the formative context of young adults.

To conclude, most of our presage variables are individual. Through this paper, we have shown the importance of other nonpersonal variables in young adults´ trajectories. Therefore, other previous relational variables should be included in future studies, such as social support, parental support, family dynamics, and relationship satisfaction, amongst others.

## 5. Conclusions

The results shown in this paper help us understand the role that individual and contextual factors have on the acquisition of psychological competencies in young adults. Self-regulation has proven to be an important meta-ability that predicts personal well-being and behavioral health outcomes. Furthermore, educational context has been shown to be an external predictor of other skills, such as problem-focused strategies, and positive outcomes, such as flourishing and health. These results nourish the idea of the importance of the combined effect of individual and contextual factors. Complementarily, other sociodemographic variables such as gender and age are crucial for the acquisition of competencies and personal strengths. College students are in a stressful developmental stage with constant changes and challenges, and because they are at a critical point, we cannot leave their growth only to free will. We must question what kind of context we are providing them and start including both factors in our research. In the academic context, interventions designed for college students must integrate the assessment of their personal self-regulation—or other meta-abilities factors that might predict positive outcomes—but also the external contextual settings they are immersed in if we want them to flourish or achieve their goals.

## Figures and Tables

**Figure 1 ijerph-19-10223-f001:**
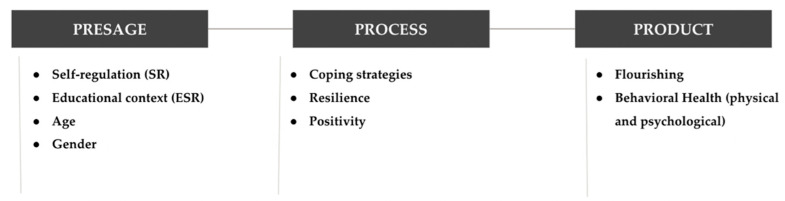
Diagram of the variables included in this study.

**Figure 2 ijerph-19-10223-f002:**
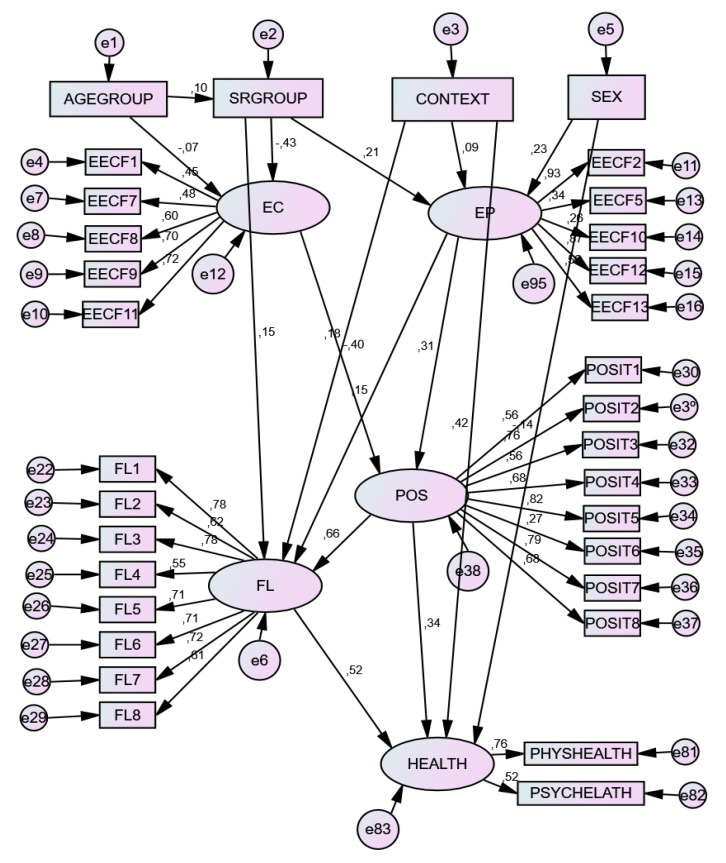
Predictive structural relationship of self-regulation level, educational context, age, and gender on coping strategies, positivity, and well-being outcomes. Note. SRGROUP = self-regulation group; CONTEXT = educational context; EC = emotion-focused coping; EP = problem-focused coping; POS = positivity; FL = flourishing; HEALTH = behavioral health; PHYSHEALTH = physical health; PSYHEALTH = psychological health.

**Table 1 ijerph-19-10223-t001:** Bivariate correlations between presage, process, and product variables.

	SR	COP	EF.COP	PF.COP	RESIL	POS	FLO	HEALTH	PHY.H	PSY.H
SR	1									
COP	0.10 **	1								
EF.COP	−0.16 ***	0.82 ***	1							
PF.COP	0.33 ***	0.84 ***	0.38 ***	1						
RESIL	0.48 ***	0.24 ***	0.01	0.39 ***	1					
POS	0.48 ***	0.16 **	−0.10 *	0.35 ***	0.58 ***	1				
FLO	0.51 ***	0.27 **	−0.1	0.50 ***	0.51 *	0.74 ***	1			
HEALTH	0.46 ***	0.07	−0.15	0.29 **	0.28	0.66 ***	0.65 ***	1		
PHY.H	0.39 ***	0.17	−0.03	0.34 ***	0.37	0.60 ***	0.65 ***	0.88 ***	1	
PSY.H	0.37 ***	−0.11	−0.28 **	0.09	0.09	0.47 ***	0.36 ***	0.75 ***	0.34 ***	1

Note. SR = self-regulation; COP = coping; EF.COP = emotion-focused coping; PF.COP = problem-focused coping; RESIL = resilience; POS = positivity; FLO = flourishing; HEALTH = health; PHY.H = physical health; PSY.H = psychological health. * *p* < 0.05; ** *p* < 0.01; *** *p* < 0.001.

**Table 2 ijerph-19-10223-t002:** Different structural linear models tested.

	X^2^	DF	*Sig.*	CMIN/DF	NFI	RFI	IFI	TLI	CFI	HOELTER *	RMSEA
M1	3,839,812	(434 − 93): 341	0.00	11,220	0.639	0.57	0.66	0.592	0.658	202	0.073
M2	3,426,504	(560 − 108): 452	0.00	7582	0.927	0.917	0.902	0.95	0.9	293	0.059

* Hoelter at 0.01.

**Table 3 ijerph-19-10223-t003:** Total, direct, and indirect standardized effects of the variables.

Predictive Variable	Criterion Variable	Total Effect	CI (95%)	Direct Effect	CI (95%)	Indirect Effect	CI (95%)	Results, Effects
SRGROUP	EF. COP	−0.435	[−0.61, −0.36]	−0.435		0.000	[−0.61, −0.36]	Direct only
SRGROUP	PF.COP	0.207	[0.822, 0.123]	0.207		0.000	[0.82, 0.12]	Direct only
SRGROUP	FL	0.339	[0.412, 0.224]	0.154	[0.067, 0.242]	0.185	[0.412, 0.225]	Partial mediation
SRGROUP	POS	0.235	[0.129, 0.376]	0.000		0.235	[0.129, 0.376]	Full mediation
SRGROUP	HEALTH	0.259	[0.123, 0.345]	0.000		0.259	[0.123, 0.345]	Full mediation
CTX	PF.COP	0.087	[0.066, 0.134]	0.087	[0.066, 0.134]	0.000		Direct only
CTX	FL	0.211	[0.102, 0.323]	0.181	[0.021, 0.222]	0.030	[0.011, 0.042]	Partial mediation
CTX	HEALTH	0.542	[0.320, 0.721]	0.422	[0.221, 0.572]	0.120	[0.081, 0.331]	Partial mediation
CTX	POS	0.027	[0.015, 0.036]	0.000		0.027	[0.015, 0.036]	Full mediation
SEX	PF.COP	0.225	[0.120, 0.312]	0.225	[0.120, 0.312]	0.000		Direct only
SEX	HEALTH	−0.078	[−0.08, −0.021]	−0.143	[−0.27, −0.08]	0.065	[0.034, 0.092]	Partial mediation
SEX	POS	0.069	[0.134, 0.051]	0.000		0.069	[0.134, 0.051]	Full mediation
SEX	FL	0.078	[0.12, 0.032]	0.000		0.078	[0.12, 0.032]	Full mediation
AGEGR	SRGROUP	0.099	[0.05, 0.13]	0.099	[0.05, 0.13]	0.000		Direct only
AGEGR	EF. COP	−0.110	[−0.06, −0.14]	−0.067		−0.043		Partial mediation
AGEGR	PF.COP	0.021	[0.010, 0.042]	0.000		0.021	[0.010, 0.042]	Full mediation
AGEGR	POS	0.050	[0.016, 0.074]	0.000		0.050	[0.016, 0.074]	Full mediation
AGEGR	FL	0.051	[0.021, 0.081]	0.000		0.051	[0.021, 0.081]	Full mediation
AGEGR	HEALTH	0.044	[0.023, 0.071]	0.000		0.044	[0.023, 0.071]	Full mediation

Note. AGEGR = age group; SRGROUP = self-regulation group; CTX = educational context; EF.COP = emotion-focused coping; PF.COP = problem-focused coping; POS = positivity; FL = flourishing; HEALTH = health; CI = confidence interval.

## Data Availability

Not applicable.

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
