# Peer review of "A Path Analysis Model of Self- vs. Educational-Context- Regulation as Combined Predictors of Well-Being in Spanish College Students"

_ijerph, 2022, doi:10.3390/ijerph191610223_

Round 1

Reviewer 1 Report

This is an excellently designed and implemented empirical research project that adds valuable data and information to the literature.  The purpose, introduction, literature review, methodology, discussion, and conclusions are presented with clarity which ensures a better understanding of the research.  The statistical methods employed to analyze the collected data.  The data collection method is very well documented, as are the survey designs.  Overall, an excellent research project.

There is one error on Line 294 which reads; 'was developed by [90].'.  The name of the developers should be shown, or the sentence altered to alleviate the requirement for the name.

Author Response

The name of the author in Line 294 was rephrased correctly. 

Thank you for the observation 

Reviewer 2 Report

This is an interesting topic and builds on a sound methodological framework. Results are coherent and further increase the knowledge on the social and personal variables affecting college students well-being. 

General comments

- Although it is perfectly understandable regarding all variables at stake, I feel the introduction section is a bit too long and detailed;

- Throughout the results section, there is no need to constantly describe where each variable belongs to. I feel it disperses the readers train-of-thought.

Specific comments:

- Across the Instruments (Materials and Methods section), there should be a coherence when expressing the validity and reliability measures (e.g. Cronbach Alpha's values are sometimes in first order and sometimes in last order);

- With so many items (126!) in stake, average response time should be mentioned;

- Not sure if I missed it but it I feel the response rate should also be mentioned;

- In the Results section, one possible strategy to overcome the repetition of variables can be to organize the results in each of the 3P's.

Typos:

line 358: remove 'was' from '(...) being the second model was the most (...)'

line 384: remove 'is' from '(...) of health in is two components (...)'

Author Response

Thank you for your observations. In the Word attached, I respond to each of the comments. 

Reviewer 3 Report

This manuscript has a solid focus and rationale. Even though the study is “exploratory” in that it means to find out a structured model and how important each of the components is.  The theories used to guide the study, and the explanations are well addressed. The data analysis procedures are good. As a reviewer, I feel this manuscript is well prepared, focused on, and consistent in presenting a story. My comments on minor revisions are as follows:

1. It could be clearer or easier to readers to grasp the key points if you use a diagram or a table or some sort of visual aid to display the relationship between the constructs in the Self- vs External Regulation Behavior Theory, the 3P Biggs Model, and the latent variables you included in your structural equation model.

In your 1.1 Theoretical Framework (page 2), you explained well how you use the 3p theory to choose the variables of study, up to line 72. You did not follow the same strategy to introduce/explain how your variables are related to the self- vs. externally regulated model.

2. Your three hypotheses on page 5 suggest this is your conceptual model of testing (see the diagram below).

Your analysis also studied the indirect effect of regulatory presage variables to product variables. 

Your description on page 8 states: “The SEM analysis was established following these theoretical statements: 1) self-regulation group and educational context (personal and contextual presage variables) would significantly and positively predict coping strategies, resilience, and positivity (process variables). 2) Coping, resilience, and positivity (process variables) would positively predict well-being constructs, namely flourishing, and health in both constructs (product variables). 3) Flourishing will positively predict health (product variables). 4) Self-regulation and educational context (personal and contextual presage variables) will positively predict flourishing and health (product variables). 5) Age and sex, which were non-latent variables, were expected to be predictors of all the latent variables studied (emotion-focused coping, problem-focused coping, positivity, flourishing, and behavioral health). Results below will be detailed according to these theoretical statements.”

My point is these three places in your manuscript could be re-addressed so they are consistent. A conceptual framework diagram may help readers to have an overview too.

3. The journal may have professional statisticians to safe-guard the statistics process. I do not object to your use of SEM. But it looks like a path analysis is sufficient to achieve your aim and test your hypotheses.  In your figure 1, and Table 3, there are many item components in your SEM model such as EECF1 to EECF11, you do not even mention. So if you focus on a path analysis the figure 1 will be much simpler. 

These above are my comments for consideration. I leave other reviewers and statistician to see if it is necessary to run a path analysis while you already did the SEM.

Thank you!

Author Response

Thank you for the suggestions. All were taken into account for the updated manuscript. In the attached file we respond to each one of the comments.  

Reviewer 4 Report

Context could have been better defined. I would like to have seen specific ideas related to improving  context as suggested from the study results. Conclusions could have been further developed.

Author Response

Thank you for your observations. They have helped to improve the manuscript. 

I agree with the limitation with the "context" variable. The changes I made in the manuscript were: 

  • I pointed it out this limitation and extended more on ways to improve it in the discussion. 
  • Conclusions were further developed, adding a few sentences to this section.